# Chromosome Instability in Pony of Esperia Breed Naturally Infected by Intestinal Strongylidae

**DOI:** 10.3390/ani12202817

**Published:** 2022-10-18

**Authors:** Emanuele D’Anza, Francesco Buono, Sara Albarella, Elisa Castaldo, Mariagiulia Pugliano, Alessandra Iannuzzi, Ilaria Cascone, Edoardo Battista, Vincenzo Peretti, Francesca Ciotola

**Affiliations:** 1Department of Veterinary Medicine and Animal Production, University of Naples Federico II, Via Delpino 1, 80137 Naples, Italy; 2National Research Council (CNR), Institute of Animal Production System in Mediterranean Environment (ISPAAM), Piazzale E. Fermi, 1, 80055 Portici, Italy; 3Independent Researcher, Via Rampa 6, 03038 Roccasecca, Italy

**Keywords:** Pony of Esperia, chromosome instability (CIN), intestinal strongylosis, eggs per gram (EPG), chromosome aberrations (CAs)

## Abstract

**Simple Summary:**

Intestinal parasites are among the main causes of hidden economic losses in livestock farming. This study reports the results of chromosome instability analyses in Esperia ponies with different intestinal strongyles fecal egg counts. Interestingly, animals with higher fecal egg counts showed increased levels of chromosome instability. If this condition is confirmed in other horse breeds and livestock species, it will be important to understand the causes in order to implement therapeutic strategies for the management of intestinal parasites.

**Abstract:**

The Pony of Esperia is an Italian autochthonous horse breed reared in the wild on the Aurunci and Ausoni Mountains. Currently, it is considered an endangered breed, as its population consists of 1623 animals. It is therefore essential to identify all aspects that can improve the management and economy of its breeding, favoring its diffusion. In this paper, the effects of intestinal strongyle infection on the chromosome stability of peripheral blood lymphocytes (PBLs) was evaluated through aneuploidy and chromosome aberration (gap, chromatid and chromosome breaks, and the number of abnormal cells) test. Statistical difference in the mean values of aneuploidy, cells with chromosome abnormalities, and chromosome and chromatid breaks were observed between ponies with high fecal egg counts (eggs per gram > 930) and those with undetectable intestinal strongylosis. The causes of this phenomenon and possible repercussions on the management of Pony of Esperia are discussed in the paper.

## 1. Introduction

The Pony of Esperia (Figure 1) is an autochthonous horse breed reared in the province of Frosinone (Lazio, Central Italy). It is characterized by a morello coat, sometimes with socks and head star, and a very thick mane and tail. The head is short and conical with a straight profile; the neck is proportionate and not excessively muscular; the robust shoulder is well-attached to the trunk; the withers is pronounced; the back is inclined; the chest is developed and muscular; the thorax is shallow; and the limbs are robust. It has a maximum withers height of 138 cm for males and 132 cm for females with a maximum live weight of 350 Kg. It originated in the area of the Aurunci and Ausoni Mountains and underwent crossbreeding along successive generations with Arabian horses of Nedjad origin. The selective pressure of the mountainous environment in which it developed determined its small size, the ability to live in hostile environments, and its rusticity. This horse was used for different purposes such as the transport of materials or people, riding, and mule production. The Pony of Esperia is currently bred as a saddle animal and for riding competitions.

This breed is reared in the wild and it stays in the pastures throughout the year. No food supplementations are administered, and pharmacological treatments are performed only when strictly necessary.

Currently, there are 1623 individuals enrolled in the studbook, so it is included in the breeds with limited diffusion to be safeguarded. For this purpose, it is necessary to improve our knowledge about their geno-morphofunctional characteristics and to highlight the conditions that make its farming more difficult and economically burdensome. The main breeding issues to which this breed is exposed are attacks by predators (which usually affect younger individuals or those in a precarious state of health), infectious diseases transmitted by other wild animals or transmitted from endo-ectoparasites. A study performed in 2006 on 230 individuals belonging to 33 families, showed the presence of polyparasitism in this breed. Intestinal Strongyles, *Parascaris* spp., *Oxyuris equi*, *Anoplocephala* spp., ticks, flies, and *Gasterophilus* spp. larvae have been found in all groups, suggesting that despite this breed being considered rustic and disease-resistant it needs parasitosis control plans [1].

Intestinal parasites are often responsible for delays in growth and the worsening of athletic performance, causing hidden economic losses in breeding [2]. Since they are often clinically asymptomatic, in livestock preventive therapies are carried out at specific times of the year, but this is not the case in Pony of Esperia breeding.

The aim of this study was to evaluate the effect of different degrees of natural intestinal strongyles infection on peripheral blood lymphocytes (PBLs) chromosome instability (CIN) in Pony of Esperia using aneuploidy and chromosome aberration (CA) tests [3]. Both tests have been used in humans and animals to evaluate the mutagenetic effects of environmental pollutants [4,5,6], drug and food supplements [7,8]; correlations among congenital malformations and chromosome stability [9,10,11], chromosome stability differences within different breeds of the same species [12], and the effects of micro- and macronutrient deficiencies [13].

PBLs are the ideal cells for CIN evaluation in an individual since they circulate throughout the body, thus being exposed to all possible risk conditions and thus representing an early and easy marker to analyze after in vitro cultivation [14].

CIN is a type of genomic instability with an increase in the numerical and structural alterations in chromosomes, and it is due to an increase in DNA damages, the malfunction of DNA damage repair mechanisms, or both [15]. The increase in CIN, as well as being a risk factor for the development of cancers, represents an index of altered homeostasis that negatively influences the well-being of the individual. Moreover, in livestock an increase in CIN has been negatively correlated to fertility and reproduction [16].

To the best authors knowledge, this is the first time that a relation between aneuploidy and CAs and the degree of intestinal parasitic infection was investigated in horse species.

## 2. Materials and Methods

### 2.1. Animals

For this study, fifty female Ponies of Esperia, aged between 3 and 20 years, were enrolled. All animals belonged to the same farm located in the province of Frosinone (Lazio Region) and were reared under the same conditions. All individuals were sampled twice (D0, before the treatment, and D14, 14 days after the treatment) for blood and feces to perform karyotype, aneuploidy, and CA tests in PBLs and fecal egg counts (FEC), respectively. At both sampling times, a clinical evaluation was performed, and the body condition scores (BCS) were determined using a five-point system (1 = poor, 2 = moderate, 3 = ideal, 4 = fat, and 5 = obese) [17] by the same investigator, and all animals were healthy, with a BCS = 3.

### 2.2. Coprological Analysis

Individual fecal samples were collected from all ponies involved in the study, and according to general recommendations proposed by Nielsen et al. [18], feces were taken directly from the rectum of each animal. Individual fecal egg counts (FECs) were performed for all ponies before the start of the trial (D0) and at 14 days post-treatment (D14) using a special modification of the McMaster method with a lower detection limit of 10 eggs per gram (EPG) using a Sheather’s saturated sugar solution with a specific gravity of 1.250 as a flotation medium [19]. Based on the morphological identification [20], each egg was classified as belonging to intestinal strongyles, *Parascaris* spp., *Strongyloides westeri*, *Oxyuris equi*, or *Anoplocephala* spp.

### 2.3. Anthelmintic Treatment

After the first collection, the ponies were divided in two groups of 25 animals that were homogeneous per age and fecal egg count: in the treatment group (T group), animals were treated with fenbendazole (FBZ) at the horse dose rate (7.5 mg/kg BW, Panacur Oral Paste, MSD Animal Health, Walton, Milton Keynes, UK), and control group (C group) animals were left untreated. Two weeks after the treatment (day 14) the blood and feces of all animals were resampled to verify changes in aneuploidy, CA, and parasitic infection. To determine the anthelmintic efficacy of FBZ, the arithmetic mean (AM) of EPG was calculated 14 days post-treatment, and the percent efficacy (%) was considered in terms of a fecal egg count reduction (FECR) test using the formula: FECR = [(AM FEC PRE-TREATMENT − AM FEC POST-TREATMENT)/AM FEC PRE-TREATMENT] × 100, according to the American Association of Equine Practitioners (AAEP) guidelines [21]. The cut-off values used to interpret the results of the FECRT were the following: efficacy > 95%, suspected resistance 90–95%, and resistance < 90% [21]. Microsoft Office Excel 2010 software was used for data recording, and FEC reductions, expressed as percentages with 95% confidence intervals, were calculated using the RESO FECRT analysis program, version 4 [22], for Excel. The simultaneous finding of a lower confidence limit (LCL) below 90% [23] and a mean percentage of FECR below 90% [21] was indicative of resistance; if only one of these two criteria was present, resistance was suspected.

### 2.4. Cytogenetic Analyses

Cell cultures for chromosome isolation were set up as reported by Ciotola et al. [24]. Briefly, peripheral blood (1 mL) was cultured at 37.5 °C in RPMI medium and enriched with fetal calf serum (10%), L-glutamine (1%), and lectin (1.5%) for 48 h. Cells were harvested after colcemid (0.3 lg/mL) treatment for 1 h and given a hypotonic treatment (KCl 0.5%) and three fixations in methanol–acetic acid (3:1), the third occurring overnight. Three drops of cell suspension were air-dried on cleaned and wet slides that were stained a day later with acridine orange (0.1% in a phosphate buffer, pH 7.0) for 3 min, washed in tap and distilled water, and mounted in the same phosphate buffer. The slides were observed 24 h after staining or later (1 week). At least 10, 100, and 50 cells per animal were examined from slides of normal cultures to perform conventional karyotype, aneuploidy, and chromosome aberration (CA) tests, i.e., gaps, chromatid and chromosome breaks, and the percentage of abnormal cells (abnormal cells are those with at least one chromatid or chromosome break) (Figure 2) [8,25]. All metaphase plates were observed under a fluorescence Nikon Eclipse 80i microscope, captured with a Nikon Sight DS-5M digital camera, transferred to a PC, and later processed with an image analysis software by two cytogeneticists.

### 2.5. Statistical Analyses

Microsoft Office Excel 2010 software was used for data recording then IBM SPSS for Windows software package version 22.0 (SPSS Inc., Chicago, IL, USA) was used for statistical analyses. The Esperia ponies were divided into five groups: group C + T included all the animals at day 0, groups C_D0_ and C_D14_ referred to the control group on day 0 and day 14, respectively, and groups T_D0_ and T_D14_ referred to the treated animals on day 0 and day 14, respectively. A statistical subanalysis was performed by dividing the groups C + T and T_D0_ according to the EPG and then according to age of the individuals (up to 6 years and equal or higher to 6 years). The parasitic burden ranges of the groups were chosen by making multiple comparisons to verify if there were ranges within which the CAs increased or decreased significantly. Age groups were established according to Wójcik and Smalec [25].

Student’s t-test was used to compare the structural percentage and cells with the CA percentage in all groups. The independent-sample *t*-test (Mann–Whitney test) was used to compare the means of the quantitative variables in the groups [26]. A Spearman correlation was performed between FEC and CAs.

## 3. Results

### 3.1. Coprological Analysis and Anthelmintic Efficacy

Intestinal strongyle eggs were found in all tested animals. At the start of the study, the mean EPG count was 992.20 ± 443.75. The mean EPG values in the T and C groups were 1096.80 ± 424.31 and 887.60 ± 446.31, respectively, and no statistical differences were observed between the two group (t = 1.6986; *p* = 0.0959). For all animals, the dose was administered carefully, and no adverse reactions were observed in any of the treated ponies. FBZ was effective in reducing FECs at 14 days after treatment, showing an FECRT = 100%.

### 3.2. Cytogenetic Analyses

All investigated animals showed a normal karyotype, thus excluding the presence of congenital numerical and structural chromosome abnormalities that could alter CIN. Thus, the identified aneuploidy and chromosomal abnormalities observed in the metaphases of the analyzed horses did not cause birth defects.

With regards to aneuploidy and the CA assay: for seven animals at D0 and another seven animals at D14, it was not possible to analyze enough metaphases.

As it has been reported that chromosomal stability in horses is influenced by age [25], to verify if this parameter significantly influenced the values of aneuploidy and CAs in the studied population, the T_D14_ group (animals negative for intestinal parasites) was divided into two groups: T_D14_ < 5 (*n* = 9) and T_D14_ > 6 (*n* = 9), comprising, respectively, individuals aged less than or equal to 5 years and individuals older than or equal to 6 years. Statistically significant differences were observed for gaps and aneuploidy (Table 1).

When considering aneuploidy, it was decreased in all groups after the first collection in a statistically significant way, while all CA values were increased in group C_D14_ and decreased in T_D14_.

When comparing groups divided according to EPG (Table 2), statistically significant differences were found for chromatid breaks, chromosome breaks, and CAs excluding gaps between groups 1T_D0_ and 2T_D0_, with all values being higher in 2T_D0_. In the comparison between 1T_D0_ and T_D14_, a statistically significant difference was found only for gaps, while in the comparison between 2T_D0_ and T_D14_ all parameters showed a statistically significant differences. Statistically significant higher mean values for all CA parameters were also observed in group 3C + T when compared to T_D14_ and for chromatid breaks, chromosome breaks, and CAs excluding gaps in 3C + T when compared to groups 1T_D0_ and 1C + T.

When animals grouped according to EPG and age (up to 5 years and older than 6 years) were compared, a statistical difference was observed only between the aneuploidy percentage and gaps of animals with high EPG (930–1970).

Finally, the Spearman correlation test showed a negative correlation between EPG and gaps (r = −0.07; *p* < 0.01) and positive correlations between EPG and chromosome breaks (r = 0.05; *p* < 0.01) and EPG and CAs excluding gaps (r = 0.33; *p* < 0.05).

## 4. Discussion

The Pony of Esperia is an endangered autochthonous breed reared in the Aurunci and Ausoni Italian Mountains. The implementation of safeguard plans for this breed is essential not only to avoid the loss of biodiversity but also to avoid the abandonment of marginal areas. For this purpose, it appears important to highlight the causes that can worsen the profitability of this breed and to verify the treatments that can improve its productivity.

One of the main causes of hidden economic losses in animal farming is parasitic infections of digestive tract. In fact, they can induce states of subclinical malnutrition due to the alteration of the intestinal absorption capacity as well as the actual subtraction of nutrients by parasites.

In this pony population, anthelmintic treatments are carried out without performing a coprological diagnosis once per year, and this anthelmintic treatment scheme is quite different from those reported in horses in Italy [27] but is very similar to that reported in the Italian donkey population [28]. Contrary to the diffusion of anthelmintic resistance worldwide [29], anthelmintic treatment with fenbendazole has been shown to be highly effective in this wild pony population, and for this reason, as reported for donkeys [30], ponies of Esperia can be considered an animal population in refugia.

With regards to the effect of intestinal strongyles on chromosome stability in PBLs, the aneuploidy and CA tests have provided interesting results.

After dividing group T_D14_ (animals with EPG = 0) into two groups based on age (up to 5 years and equal to or older than 6 years), a statistically significant difference was observed between aneuploidy and gap. It is interesting to note that, while the gaps were higher in younger ponies, a reduction was observed in the older ones in favor of the percentage of aneuploid cells that instead was higher in older animals. This result is congruent with what was observed by Wojcik and Smalec [25], who found that in horses the mean value of SCEs was significantly different between horses under 6 years of age and horses older than 6 years, indicating that the age of the investigated animals also represents a reference parameter for sampling when studying CIN by aneuploidy and CAs but only when considering gaps. In fact, abnormal cells percentages, chromatid and chromosome breaks, and CAs excluding gap are not affected by age.

The main finding is that the mean values of chromosome breaks and CAs excluding gaps significantly increase in ponies with high fecal egg counts. Another interesting result is that a statistically significant higher value of CAs was found in animals with EPG > 930 when compared to animals with EPG < 930. In this regard, it has been seen that, in equids, the number of parasites present could be decisive in evaluating the state of health of an individual, for which EPG < 500 represents a low infection, 500 < EPG < 1000 represents a moderate infection, and EPG > 1000 represents a high infection [31,32]. The only parameter in which an inconsistency was observed is aneuploidy, which was higher in subjects with 260 < EPG < 640 than in animals with 690 < EPG < 930. However, the differences were not statistically significant.

When animals were grouped by EPG and then by age, statistically significant differences were only observed in the mean number of gaps in individuals with high EPG (950–1970).

The data reported here show that intestinal parasites affect chromosomal stability as evaluated by aneuploidy and CAs. The studies published to date on the application of chromosomal stability tests in different animal species show correlations with age and breed, while correlations with any parasitic condition have never been considered. This study is therefore the first to report this effect, but the data are limited to aneuploidy and CA tests. It therefore appears necessary to verify whether other tests, such as SCE, micronuclei, the comet assay, and fragile sites, can also be affected by intestinal parasites or other types of endoparasitosis.

The CA test detects exposure to substances that cause DNA strand breaks or a deficiency of substances that are involved in DNA repair mechanisms but does not allow the identification of the clastogen class or micronutrient deficiency [33]. Thus, it is possible to hypothesize that increased CAs in animals with high EPG could be due to a subclinical deficiency of micro- and macronutrients linked to the action of parasites present in the intestine, which in addition to subtracting nutrients, cause alterations in the composition of the microbiota [34] (fundamental in these species to produce micro- and macronutrients) and to the structure of the intestinal barrier, with consequences for the absorption ability. Another possibility is that the alterations in the intestinal barrier of the microbiota and the inflammatory processes induced by parasites allow the passage of toxic substances into the circulation that would normally be eliminated with the feces and that have a negative effect on PBLs. Both situations may cause decreases in sport performance and fertility, which in both cases, become possible causes of reduced earnings with significant impacts on the profitability of the breeding of native breeds such as the Pony of Esperia.

These hypotheses, if confirmed through specific dosage analyses of micro- and macronutrients or of other substances at the serum level, would suggest the intervention, in subjects with EPG > 930, with appropriate food supplements. Moreover, it would be interesting to evaluate in other horse breeds and in other livestock species the impact on CIN of intestinal strongyle FEC, verifying the minimum EPG count at which it is appropriate to carry out pharmacological treatments, possibly associated with food supplement administration. These data would be useful for breeds raised in the wild, for the semi-extensive technique in which carrying out pharmacological treatments is not particularly easy, and for animals in intensive farming in which the improper use of pharmacological molecules causes a considerable environmental impact. Furthermore, in native breeds these data acquire further value since the low profitability of breeding involves an accurate evaluation of the cost/benefit ratios of any treatment that is selected.

## 5. Conclusions

According to the results observed in this study, it is possible to speculate that in the wildly reared Pony of Esperia breed: (1) an anthelmintic treatment with fenbendazole is highly effective, (2) a high FEC (>930 EPG) causes an increase in CAs that is reduced after only 14 days of drug treatment, and (3) in a cost–benefit assessment, the treatment of parasitosis could be useful in subjects with FEC > 930. Finally, an important aspect emerges that should be verified with future studies: individuals with FEC > 930 could benefit from food supplementation with micro- and macronutrients.

## Figures and Tables

**Figure 1 animals-12-02817-f001:**
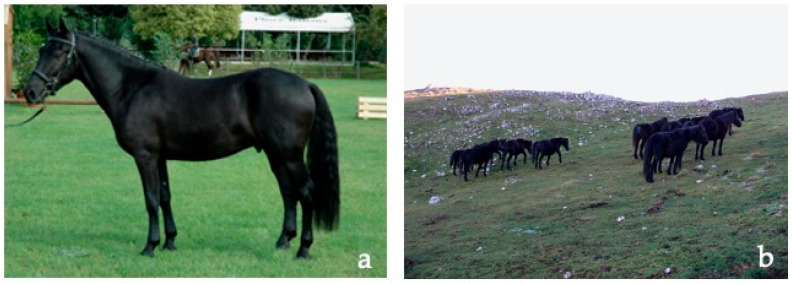
(**a**) Pony of Esperia. (**b**) Grazing herd of Pony of Esperia.

**Figure 2 animals-12-02817-f002:**
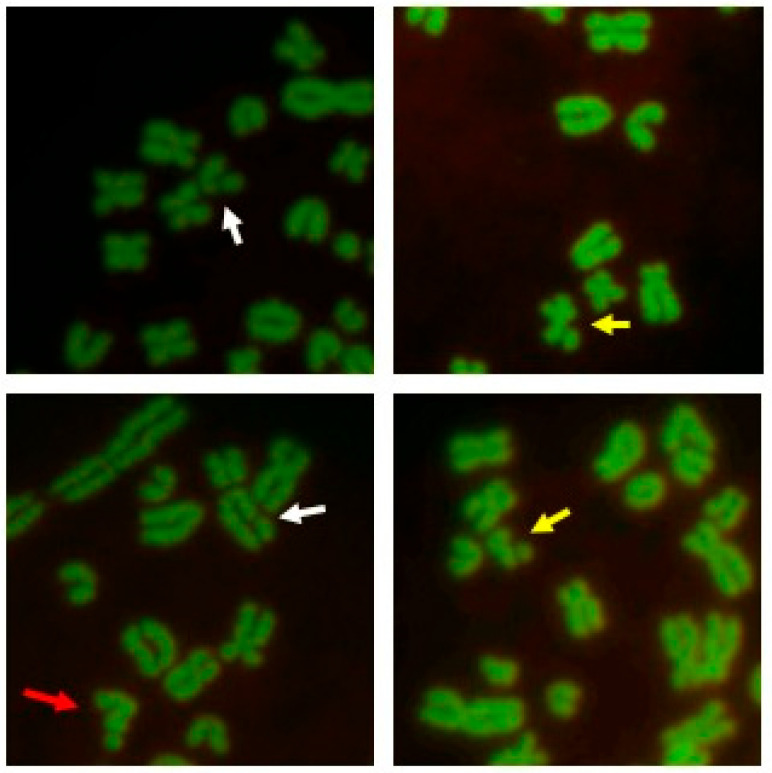
Details of chromosome metaphase plates of Pony of Esperia. The white arrow indicates a gap, the yellow arrow indicates a chromatid break, and the red arrow indicates a chromosome break.

**Table 1 animals-12-02817-t001:** Percentages of aneuploidy and abnormal cells and mean values and standard deviations of CA in groups C + T (all animals before the treatment), C_D0_ (untreated animals at T0), T_D0_ (treated animals before the treatment), C_D14_ (untreated animals at T14), T_D14_ (treated animals at T14), T_D14_ < 6 (treated animals at T14 up to 6 years), and T_D14_ > 6 (treated animals at T14 older than 6 years).

Group	N of Animals	Age		EPG		Aneuploidy (2n ≠ 64)	Abnormal Cells	Gaps	Chromatid Breaks	Chromosome Breaks	CAs Excluding Gaps
		Mean ± SD	Range	Mean ± SD	Range	%	%	Mean ± SD	Mean ± SD	Mean ± SD	Mean ± SD
C + T	43	7.63 ± 5.02	3–20	980.24 ± 471.75	260–1970	16.76 ^a,A^	6.33	0.71 ± 0.93	0.06 ± 0.25 ^a^	0.01 ± 0.11 ^a^	0.07 ± 0.28 ^A^
C_D0_	20	8.00 ± 6.04	3–20	904.44 ± 536.34	260–1970	15.44 ^a,b,A^	4.70	0.72 ± 0.98	0.04 ± 0.22 ^a,A^	0.01 ± 0.10	0.05 ± 0.25 ^a^
T_D0_	23	7.45 ± 4.06	3–20	1039.57 ± 389.47	400–1970	17.90 ^a,A^	7.74	0.70 ± 0.88	0.07 ± 0.27 ^B^	0.01 ± 0.12	0.08 ± 0.30 ^A,b^
C_D14_	13	7.75 ± 6.25	3–20	1179.00 ± 801.82	110–2760	11.30 ^c,B^	7.20	0.79 ± 0.91 ^A^	0.07 ± 0.28 ^B^	0.01 ± 0.12	0.08 ± 0.31 ^A,b^
T_D14_	23	7.45 ± 4.06	3–20	0	0	13.32 ^b,c^	6.64	0.61 ± 0.74	0.04 ± 0.19 ^b,A^	0.00 ± 0.07 ^b^	0.04 ± 0.2 ^B^
T_D14_ < 6	12	3.91 ± 0.95	3–5	0	0	11.25 ^c,B^	6.31	0.77 ± 0.80 ^A^	0.04 ± 0.18 ^b,A^	0.00 ± 0.00 ^a^	0.04 ± 0.18 ^B^
T_D14_ > 6	11	10.25 ± 4.91	6–20	0	0	16.05 ^b,A^	7.45	0.49 ± 0.66 ^B^	0.04 ± 0.21 ^b,A^	0.00 ± 0.06 ^a^	0.05 ± 0.22 ^a^

^a,b,c^ = *p* < 0.05; ^A,B^ = *p* < 0.005.

**Table 2 animals-12-02817-t002:** Percentages of aneuploidy and abnormal cells and mean values and standard deviations of chromosome abnormalities in ponies grouped according to EPG and age at D0 and compared with the T_D14_ group.

Group	N of Animals	Age		EPG Range	Aneuploidy (2n ≠ 64)	Abnormal Cells	Gaps	Chromatid Breaks	Chromosome Breaks	CAs Excluding Gaps
		Mean	Range		%	%	Mean ± SD	Mean ± SD	Mean ± SD	Mean ± SD
1T_D0_	11	7.08 ± 4.32	3–16	400–800	17.39	5.69	0.87 ± 0.88 ^A^	0.05 ± 0.23 ^A^	0.00 ± 0.07 ^A^	0.05 ± 0.31 ^A,a^
2T_D0_	9	8.23 ± 5.43	3–20	950–1970	14.47	10.22	0.81 ± 0.86 ^A^	0.08 ± 0.29 ^B^	0.03 ± 0.16 ^B^	0.11 ± 0.33 ^B^
1C + T	12	8.4 ± 6.12	3–20	260–640	18.62	5.20	0.74 ± 0.96	0.05 ± 0.23 ^A^	0.00 ± 0.06 ^A^	0.05 ± 0.24 ^A,a^
2C + T	13	6.17 ± 3.24	3–12	690–930	12.26 ^a^	6.92	0.76 ± 0.94	0.07 ± 0.27	0.01 ± 0.10	0.08 ± 0.30 ^b^
3C + T	18	9.00 ± 6.14	3–20	950–1970	17.85	10.89	0.81 ± 0.88 ^A^	0.08 ± 0.30 ^B^	0.04 ± 0.19 ^B^	0.12 ± 0.35 ^B^
1–2C + T < 6	12	3.82 ± 0.94	3–5	260–930	19.13 ^b^	10.00	0.76 ± 0.94	0.06 ± 0.27	0.01 ± 0.09	0.07 ± 0.29
1–2C + T > 6	11	10.24 ± 4.15	6–20	260–930	18.41	9.64	0.80 ± 0.95	0.06 ± 0.25	0.01 ± 0.08	0.07 ± 0.27
3C + T < 6	9	3.40 ± 0.80	3–5	950–1970	12.30 ^a^	10.64	1.12 ± 0.98 ^A^	0.06 ± 0.26	0.05 ± 0.21 ^B^	0.11 ± 0.34 ^B^
3C + T > 6	9	11.00 ± 4.79	6–20	950–1970	16.52	11.67	0.72 ± 0.83 ^B^	0.08 ± 0.31 ^B^	0.03 ± 0.17 ^B^	0.11 ± 0.34 ^B^
T_D14_	23	7.45 ± 4.06	3–20	0	13.32	6.64	0.61 ± 0.74 ^B^	0.04 ± 0.19 ^A^	0.00 ± 0.07 ^A^	0.04 ± 0.21 ^A,a^

^a,b^ = *p* < 0.05; ^A,B^ = *p* < 0.005.

## Data Availability

The data presented in this study are available on request from the corresponding author.

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
