# Peer review of "Chromosome Instability in Pony of Esperia Breed Naturally Infected by Intestinal Strongylidae"

_animals, 2022, doi:10.3390/ani12202817_

Round 1
Reviewer 1 Report
The authors find that the lymphoid cells of sick pony have a higher rate of chromosome instability (aneuploidy, CA). Given that the sick animals have all been diagnosed intenstinal strongylidae. These assays can therefore serve as a potential biomarker for these populations but work needs to be improved:
1. The Authors write that they analyzed 50 female Ponies of Esperia aged 3 to 20 years. The Authors should divide the animals into two age groups because age significantly influences the level of damage to the genetic material. Therefore, the tables should be corrected according to age.
2. The Authors checked the correlation between CA and FEC. Why was the correlation between the remaining analyzed instabilities not checked?
3. The Authors did not include photos with identified chromosomal instabilities, which are irrefutable evidence of the tests. Instead of two photos of horses (one photo is enough), you should place metaphase plates from the chromosome staining techniques used, with marked instabilities.
4. In line 167-169 the Authors write that in all the tested animals a normal diploid karyotype was found. Next, they state the aneuploidy, which are included in tables 1 and 2. I think the sentence in line 167 - 169 would be more appropriate: The identified aneuploidy and chromosomal abnormalities observed in the karyotypes of the analyzed horses did not cause birth defects.
5. In line 170-172, the Authors wrote that not all animals in each group were cytogenetically analyzed due to insufficient metaphases. To avoid such situations, researchers should in the future establish duplicate in vitro breeding from the animal for future cytogenetic testing.
6. In line 185-186 change the description of the correlation from (0.05; p<0.01) to (r=0.05; p<0.01) and (0.33; p<0.05) to (r=0.33; p<0.05)
7. In the tables, the authors give different levels of significance, for example p <0.05, p <0.001, p <0.005, p <0.01. At what significance levels were the differences between the mean verified? This is usually done for p<0.01 and p<0.05. Verification should be standardized for all analyzes.
8. In line 241-242 the authors write about the possibility of using the minimum dose of fenbendazole. Thanks to their research, according to veterinary data, the number of EPGs is more than 200 strongylidae eggs to apply fenbendazole. Only horses that shed this amount of eggs in their feces should be counted. This allows you to reduce the cost of deworming, promotes the effectiveness of deworming preparations and prevents resistance.
9. The authors should slightly expand the discussion about the negative influence of the investigated mutagenic factors on the equine karyotype of other scientists.
10. The statement in the sentence in line 253-254 is "far-reaching" and may be made in subsequent possibly subsequent studies, but not confirmed in the present research.
After considering my comments, the work can be published.
Author Response
Reviewer 1 report 2022/10/05
Dear Reviewer,
all the authors wish to thank you for the useful suggestions aimed at strengthening the scientific value of this study. Below are the point-by-point answers:
The authors find that the lymphoid cells of sick pony have a higher rate of chromosome instability (aneuploidy, CA). Given that the sick animals have all been diagnosed intenstinal strongylidae. These assays can therefore serve as a potential biomarker for these population but work needs to be improved:
- The Authors write that they analyzed 50 female Ponies of Esperia aged 3 to 20 years. The Authors should divide the animals into two age groups because age significantly influences the level of damage to the genetic material. Therefore, the tables should be corrected according to age.
The tables were implemented by also analyzing the age-related data in the different groups. We used 6 years as the cutoff as the only reference we found in the Horse is the paper:
Wójcik E, Smalec E. The effect of environmental factors on sister chromatid exchange incidence in domestic horse (Equus caballus) chromosomes. Folia Biol (Krakow). 2013 61(3-4):199-204. doi: 10.3409/fb61_3-4.199.
- The Authors checked the correlation between CA and FEC. Why was the correlation between the remaining analyzed instabilities not checked?
We analyzed the correlations for all the parameters, but the only ones were those reported in the work and a negative correlation between gap and FEC. We have modified the text trying to clarify this aspect better (see: 202-204)
- The Authors did not include photos with identified chromosomal instabilities, which are irrefutable evidence of the tests. Instead of two photos of horses (one photo is enough), you should place metaphase plates from the chromosome staining techniques used, with marked instabilities.
It has been added a collage of images in which the different types of CAs found are showed (see Fig. 2)
- In line 167-169 the Authors write that in all the tested animals a normal diploid karyotype was found. Next, they state the aneuploidy, which are included in tables 1 and 2. I think the sentence in line 167 - 169 would be more appropriate: The identified aneuploidy and chromosomal abnormalities observed in the karyotypes of the analyzed horses did not cause birth defects.
The text has been changed according to your suggestion to better clarify the condition (See 176-178).
- In line 170-172, the Authors wrote that not all animals in each group were cytogenetically analyzed due to insufficient metaphases. To avoid such situations, researchers should in the future establish duplicate in vitro breeding from the animal for future cytogenetic testing.
Normally a duplicate experiment in vitro is carried auto each animal, but it is based on the availability of sampled biological material (blood). Since the Pony of Esperia is bred in the wild, in some cases it was difficult to sample enough blood to set up cell cultures in duplicate.
- In line 185-186 change the description of the correlation from (0.05; p<0.01) to (r=0.05; p<0.01) and (0.33; p<0.05) to (r=0.33; p<0.05)
Thanks for the suggestion, the MS has been modified (see: 202-204).
- In the tables, the authors give different levels of significance, for example p <0.05, p <0.001, p <0.005, p <0.01. At what significance levels were the differences between the mean verified? This is usually done for p<0.01 and p<0.05. Verification should be standardized for all analyzes.
We have standardized the significance in the two tables.
- In line 241-242 the authors write about the possibility of using the minimum dose of fenbendazole. Thanks to their research, according to veterinary data, the number of EPGs is more than 200 strongylidae eggs to apply fenbendazole. Only horses that shed this amount of eggs in their feces should be counted. This allows you to reduce the cost of deworming, promotes the effectiveness of deworming preparations and prevents resistance.
Thanks for the observation, this is precisely one of the aspects we wanted to evaluate with this study!
- The authors should slightly expand the discussion about the negative influence of the investigated mutagenic factors on the equine karyotype of other scientists.
There are few studies on CIN in the equine species and typically they have been done by applying other chromosome stability tests. Surely this work highlights that intestinal parasites can influence chromosomal stability, and it would be necessary to verify how much this condition influences the results obtainable with other tests (SCE, Micronuclei, Comet assay ...). It should also be considered, in general, in chromosomal stability studies conducted on animals. We tried to clarify this aspect in the paper (see: 254-261)
- The statement in the sentence in line 253-254 is "far-reaching" and may be made in subsequent possibly subsequent studies, but not confirmed in the present research.
We thank the reviewer for this observation, we had in fact written in the discussions (see:264-279) that this aspect should be investigated with future studies, but we have not clarified in the same way in the conclusions. We have therefore changed the conclusions (see:294-296).
After considering my comments, the work can be published.
We have tried to fulfill all your requests and we hope that the work is now suitable for publication.
Best regards
Sara Albarella

Reviewer 2 Report
The study "Chromosome instability in Pony of Esperia breed naturally infected with intestinal strongylidae" by D'Anza et al. made a lasting impression on me. It is a very significant, carefully planned, and researched effort.
The study's objectives, methodology, and findings are all regularly presented in the abstract section and are all adequate for the study.
The introduction is understandable and clear. Although the content and methodology are nicely stated, it would be preferable if the study's limitations were included. Additionally, parasite names must be italicized (line 108). Panacur should have a city and a country added (line 114).
The tables and presentation of the findings in the results section are satisfactory. There is a typo that has to be fixed (paragraphers, line 181).
References, a discussion, and a conclusion are sufficient. At Cas, there are mistakes (lines 215 and 251). Additionally, the language is highly fluid and simple to understand. I advise minor revision.
Author Response
Reviewer 2 report 2022/10/05
Dear Reviewer,
we wish to thank you for the comments. We have modified the paper according to your indications.
The study "Chromosome instability in Pony of Esperia breed naturally infected with intestinal strongylidae" by D'Anza et al. made a lasting impression on me. It is a very significant, carefully planned, and researched effort.
The study's objectives, methodology, and findings are all regularly presented in the abstract section and are all adequate for the study.
The introduction is understandable and clear. Although the content and methodology are nicely stated, it would be preferable if the study's limitations were included.
We have tried to clarify the limitations of the study in the discussions (see 255-261)
Additionally, parasite names must be italicized (line 108). Panacur should have a city and a country added (line 114).
Modified (see 108 and 114)
The tables and presentation of the findings in the results section are satisfactory. There is a typo that has to be fixed (paragraphers, line 181).
Modified (see 196)
References, a discussion, and a conclusion are sufficient. At CSs, there are mistakes (lines 215 and 251). Additionally, the language is highly fluid and simple to understand. I advise minor revision.
We hope that the changes correspond to your request and that the article is now suitable for publication
Best regards
Sara Albarella

Round 2
Reviewer 1 Report
After I read the revised work by Authors, I state that all my comments and sugestions have been made. Therefore, I recommend these works to the journal Animals.